# Perception, Knowledge, and Consumption Potential of Crude and Refined Palm Oil in Brazilian Regions

**DOI:** 10.3390/foods13182923

**Published:** 2024-09-15

**Authors:** Alana Moreira Bispo, Agnes Sophia Braga Alves, Edilene Ferreira da Silva, Fernanda Doring Krumreich, Itaciara Larroza Nunes, Camila Duarte Ferreira Ribeiro

**Affiliations:** 1Nutrition School, Federal University of Bahia, Basílio da Gama Street, w/n-Campus Canela, Salvador 40110-907, Brazil; alanamoreira.bispo@gmail.com (A.M.B.); fernandakrumreich@ufba.br (F.D.K.); 2Graduate Program in Food Science, Faculty of Pharmacy, Federal University of Bahia, Campus Ondina, Salvador 40170-290, Brazil; asbasophia@hotmail.com; 3Graduate Program in Food Science, Department of Food Science and Technology, Federal University of Santa Catarina, Admar Gonzaga Highway, 1346, Itacorubi, Florianópolis 88034-000, Brazil; edilenesilva1744@gmail.com (E.F.d.S.); itaciara@yahoo.com (I.L.N.)

**Keywords:** palm oil, *Elaeis guineensis*, online research, carotenoids, vitamin E, nutrition, health

## Abstract

Crude palm oil (CPO) is the most produced vegetable oil globally, with Brazil contributing only 0.74% of global production. Pará and Bahia account for more than 82% of Brazil’s output. Despite its widespread use in the food industry after refining, there is little research on CPO consumption and perception in Brazil, particularly regarding its nutritional aspects. This study, conducted between March and July 2022, explored Brazilians’ perceptions and the potential for CPO consumption. The results show that most participants are unfamiliar with CPO but view its nutrients favorably. Less than half regularly purchase CPO. Refined palm oil (RPO) is even less known, with many unaware that refining CPO can produce carcinogenic substances. The respondents showed little concern about RPO in their foods, rarely noticing its presence on labels. Despite limited knowledge, participants understand that refining reduces CPO’s health benefits, leading to a greater preference for crude oil over refined oil. This study highlights the need for better dissemination of information about CPO in Brazil, emphasizing its nutritional benefits and the importance of adhering to daily lipid intake limits. Adding CPO at the end of cooking or consuming it raw to preserve thermosensitive compounds is also recommended.

## 1. Introduction

Crude palm oil (CPO), extracted from the mesocarp of the fruit (palm) of the African species *Elaeis guineensis* (oil palm), is the most produced, exported, and globally imported vegetable oil, according to the United States Department of Agriculture (USDA) [1], with a production of 78.22 million tons in 2022–2023.

Due to its composition and high productivity, palm oil production is expected to meet approximately 36% of the world’s demand for vegetable oils by 2031, which is estimated to be 249 million tons [2]. A study conducted [3] established a maximum estimate of 340 million tons of vegetable oil by 2050, with the author suggesting that palm oil could supply more than half of this demand.

Southeast Asia leads the global production of this oil, with Indonesia, Malaysia, and Thailand being the most prominent producers [1]. Brazil ranks tenth in this production, with the states of Pará and Bahia concentrating most of the national CPO production (USDA, 2023b) [4]. This crude oil is widely used in culinary preparations in the state of Bahia, which is located in the country’s northeastern region [5].

CPO is an important source of carotenoids and vitamin E [2]. This is attributed to the fact that its raw material, the oil palm fruit, is recognized as the richest plant source of these nutrients [6]. Carotenoids are tetraterpenoid compounds with yellow, orange, or red coloration, some of which, such as alpha-carotene, beta-carotene, and beta-cryptoxanthin, can be converted into vitamin A in the human body [7].

This oil has two fractions: a liquid one (olein or “flor do dendê” as known in Brazil) containing unsaturated fatty acids (predominantly oleic and linoleic acids) [8] and a solid fraction (stearin or “bambá” as referred to by Brazilians) that tends to solidify at room temperature [9]. The solid fraction is characterized by saturated fatty acids, predominantly palmitic acid [8,10].

Due to its strong odor and distinctive flavor [11], this oil is predominantly consumed in its refined form, allowing for its widespread application in various food industry products (French fries, margarine, confectionery products, dairy substitutes, pizzas, etc.). As a result, approximately 80% of the entire production of crude palm oil is directed towards this refining process [6], which leads to the loss of certain nutrients, such as vitamin E (from 41.7% to 55%) and carotenoids (from 2 to 29%) [8]. During the refining process, there is also the potential formation of substances with carcinogenic risk to humans—chloropropanols such as 3-monochloro-1,2-propanediol (3-MCPD), 2-MCPD, and glycidol [12,13,14].

Wassmann, Siegrist, and Hartmann [15] assessed Swiss consumers’ perception of refined palm oil (RPO). Their study found predominantly negative associations, with RPO being viewed more negatively than other oils and fats and not widely accepted in many products. The negative perceptions were mainly related to sustainability issues.

In evaluating consumer perception of RPO through recent news observations and a survey of Spanish and Peruvian consumers, ref. [16] noted that RPO is considered detrimental to health in Spain, while Peruvians believe the opposite. Despite this, studies in Brazil are needed to investigate consumers’ perception, knowledge, and consumption potential from a nutritional perspective regarding these two types of oil from the *Elaeis guineensis* species. Thus, this study is pioneering in the country and could contribute to a better understanding of Brazilian consumers regarding these oils. This study aimed to comprehend the perception, knowledge, and consumption potential of crude and refined palm oil in Brazilian regions.

## 2. Materials and Methods

### 2.1. Participants

This study recruited participants through email, social media, and messaging apps (Instagram version 105.0.0.18.119 and WhatsApp version 2.22.25.13). Inclusion criteria were residents of Brazil, 18 years or older, with Internet access, who agreed to the informed consent terms (n = 1082). A total of 17 participants were excluded for incorrect answers to a verification question, resulting in 1065 participants. The Federal University of Bahia’s Ethics Committee, Brazil, approved this study (approval no. 5.306.483; CAAE no. 56357822.1.0000.5023) on 23 March 2022. Data were collected from March to July 2022 using the Google Form survey platform. 

### 2.2. Data Production Procedure

The target audience’s perception, knowledge, and consumption potential were analyzed through qualitative and quantitative methods. This involved assessing perception, cognition, preference, attitude, and frequency, totaling 35 questions. The questions were distributed as follows: 7 general questions regarding socioeconomic aspects; 27 questions for analyzing the perception, knowledge, cognition, preference, and consumption potential of crude and refined palm oils; and 1 verification question (Appendix A).

Table 1 provides an overview of the questions addressed in the survey. A 5-point Likert scale was utilized for the assessment of perception, attitude, frequency, and consumption potential, ranging from strongly agree, partially agree, neither agree nor disagree, partially disagree, to strongly disagree; unhealthy, somewhat unhealthy, neutral, moderately healthy to healthy; never, semiannually, monthly, biweekly, daily; and none, a little, neutral, a lot to a great deal.

The confidentiality and anonymity of the data were ensured, as participant names were not requested, and responses were evaluated by the research tool without human interference.

### 2.3. Statistical Treatment

The data were expressed in percentages and absolute frequencies and presented in tables, graphs, and figures. Correspondence analysis was employed to identify associations among categories related to the participant’s region of residence, income, types of oils consumed, cognition regarding palm oil types, and the potential consumption of foods containing these types of oil. The data were statistically analyzed using STATISTICA version 7.0 software.

## 3. Results

### 3.1. Consumer Profile

This study involved 1065 participants from Brazil, with the majority residing in the northeastern region (40.37%). The participants were predominantly female (69.76%), with an age range of 30–39 years (27.51%) and living alone (12.68%). Among them, 63% had postgraduate degrees, worked fulltime (61.60%), and had an income of over nine minimum wages (35.87%) (Table 2).

### 3.2. Consumption of Edible Vegetable Oils in Brazilian Regions

Appendix A illustrates the respondents’ five most consumed vegetable oils, focusing on olive oil (89.67%) and soybean oil (65.63%). Other oils, such as sesame oil (5.91%) and refined palm oil (3.1%), were the least consumed (Appendix A).

When evaluating the reasons for choosing a specific type of oil, price was the most significant factor for 55.77% of the participants, while taste influenced 55.21% of the choices (Appendix A).

Figure 1 presents the correlation analysis based on the type of oil most consumed by respondents (soybean oil (SO), palm oil (RPO), olive oil (OO), palm kernel oil or crude palm oil (CPO), canola oil (RO), corn oil (CO), sunflower oil (SFO), sesame oil (SEO), coconut oil (CNO)), the region of residence (north, northeast, south, midwest, southeast), and their income level (less than 1 minimum wage, between 1 and 3 minimum wages, between 3 and 6 minimum wages, between 6 and 9 minimum wages, more than 9 minimum wages, prefer not to disclose), consisting of two dimensions (dimension 1 and dimension 2, demonstrating 84.30% data correlation). The analysis revealed a significant correlation between minimum wage income and soybean and coconut oil consumption in the midwest and north regions. On the other hand, the consumption of olive oil, sunflower oil, and corn oil, which generally have higher prices, is associated with income above six minimum wages and residence in the southeast region (Figure 1).

### 3.3. CPO

#### 3.3.1. Perception and Knowledge about CPO

When participants were asked if they had heard of CPO, the majority (53%) responded negatively. Only 39.5% of the respondents knew that CPO refers to the popular palm oil. Less than 5% (4.13%) thought CPO could be animal fat, olive oil, sweet oil, soybean oil, corn oil, sunflower oil, or canola oil. Among those surveyed, 70.9% stated that they liked the taste of CPO (Appendix A).

In the survey, 63.66% of participants believe CPO is a source of lipids, 56.06% of provitamin A carotenoids, and 40.75% of vitamin E. At least one of the nutrients, such as vitamin C, vitamin K, proteins, and carbohydrates, was indicated by 59.15% of the people who took part in the survey (Appendix A).

Appendix A displays statements presented to respondents regarding CPO for them to express their agreement. Regarding the first statement, which discusses CPO being the most produced in the world, 77.93% either did not have an established opinion or expressed disagreement. Regarding the second statement concerning the study of its antioxidant substances, most respondents agreed to some extent (53.99%), but a significant portion neither agreed nor disagreed (41.50%). It was observed that most people agreed with all the other statements about CPO. That is, respondents believe that CPO has two phases (liquid and solid) (60.00%), is used in the preparation of dishes for African matrix religious practices (80.00%) and Catholic practices (Holy Week and the Day of Saint Cosmas and Damian) (72.39%), and are concerned about whether production is associated with deforestation and loss of biodiversity (63.85%).

#### 3.3.2. Acquisition of CPO

Respondents from different regions had varying perceptions regarding the “purchase” or “nonpurchase” of CPO. The only region where the proportion of purchase is higher than that of nonpurchase for CPO is the northeast (58.60%) (Appendix A), as evidenced in the chi-square analysis, where only the northeast showed responses with greater significance (*p* < 0.05) for the “purchase” of crude palm oil. The central–west and south regions showed less significance for “purchase” (Table 3). Responses from participants in the north, southeast, and northeast regions were less significant (*p* < 0.05) for the “nonpurchase” option, while the south showed greater significance for “nonpurchase” (*p* < 0.05). 

In general, less than half of the respondents usually buy CPO (42.16%). Among them, the majority (59.46%) purchase less than 500 mL of the oil, and 69.04% of people buy it semiannually, with most (75.05%) acquiring this oil from supermarkets and 47.44% obtaining it from markets (Appendix A). Regarding the presence of labels on products, only 14.25% of respondents reported buying oil without labeling. The leading information observed on the label was the date of manufacture, expiration date, lot number (73.50%), brand name (48.33%), and ingredients (27.84%).

Regarding how CPO is encountered, 45.21% of people find it exposed to light, and 87.97% of respondents reported encountering the oil at room temperature. Additionally, 78.39% purchase it in transparent plastic bottles, and 56.35% of people buy it in transparent glass bottles. Moreover, only 18.93% of those who buy CPO typically store it in the refrigerator, while the majority (73.27%) mention keeping the oil in the cupboard (Appendix A).

The survey revealed that most people identify oil rancidity by its odor and texture, and out of the total respondents, 31.85% reported discarding the oil in the trash (Appendix A).

#### 3.3.3. Use of CPO in Cooking

Among the two phases of CPO (olein and stearin), 43.5% of individuals responded that they find it better to cook with a mixture of both phases. Among the respondents, 33.90% said they chose the phases for better taste, and 27.51% of people chose for better nutritional quality. Moreover, when asked how they use/would use CPO, 76.90% responded that it would be for cooked preparations, such as moquecas, caruru, vatapá, or abará, and 58.50% use/would use it for frying, like acarajé, fish, or dumplings.

#### 3.3.4. Consumption of Foods with CPO

In the present study, on average, 57.6% of people responded that they have never consumed most foods with CPO, with xinxim de bofe, xinxim de galinha, and quiabada with palm oil being the least consumed foods (86.85%, 75.12%, and 73.80% responding “never”, respectively). Regarding the most consumed dishes, moqueca ranks first (72.77%), followed by acarajé (68.64%), vatapá (59.44%), and bobó de camarão (58.97%). Bobó de camarão is the most consumed semiannually (47.32%). Moqueca and acarajé are consumed more frequently semiannually (45.63% and 46.38%, respectively), monthly (21.13% and 16.90%, respectively), and biweekly (5.73% and 4.98%, respectively). Vatapá is the third most consumed monthly (16.24%) and biweekly (4.69%). Other dishes are more popular in daily consumption, such as fish fried in palm oil, farofa with palm oil, and pirão with palm oil (0.94%, 0.94%, and 0.56%, respectively) (Appendix A). Appendix A describes foods prepared with CPO and their main ingredients. A majority of people reported consuming beverages alongside foods with CPO (82.57%), with the main choices being soda (19.72%), followed by water (18.95%), juice (18.60%), and beer (18.24%).

### 3.4. RPO

#### 3.4.1. Knowledge about RPO

Appendix A illustrates the statements about RPO presented to respondents to indicate their agreement. Although options 1, 2, and 5 were true regarding the processing or use of RPO in the food industry, it was observed that the majority of respondents do not have a definite opinion, with “neither agree nor disagree” response percentages of 79.62%, 67.89%, and 55.87%, respectively. They also remained impartial regarding whether a product contains RPO, as seen in the third option (63.94%). In the fourth statement, 69.48% of respondents disagreed with the statement, indicating that most participants in this study do not check whether refined palm oil is listed in the ingredients (Appendix A).

#### 3.4.2. Consumption of Foods with RPO 

In Appendix A, the most consumed foods with RPO by respondents and their highest consumption frequencies are summarized. Among the analyzed foods, margarine, tomato sauce, and granola are the most consumed, with higher percentages of “daily” responses (20%, 17.18%, and 13.43%, respectively). Generally, when consumed, most mentioned foods are consumed semiannually, except for margarine (daily), tomato sauce (biweekly), chocolate bars, and ice cream (both monthly). On the other hand, the least consumed foods are instant noodles, breakfast cereals, and protein bars, with higher percentages of “never” responses (52.21%, 51.64%, and 43.19%, respectively).

### 3.5. Cognition Regarding CPO and RPO

Respondents’ cognition regarding CPO and RPO was investigated based on their opinions on how healthy these oils are for consumers. Out of the total respondents, 615 individuals (57.75%) consider CPO moderately healthy or healthy (Figure 2A), while only 354 individuals (33%) consider RPO in this way (Figure 2B).

Figure 3 represents correspondence analyses based on responses regarding how healthy palm oil types are perceived by respondents (healthy, moderately healthy, neutral, slightly healthy, unhealthy) and their region of residence (north, northeast, south, midwest, southeast). It consists of two dimensions (dimension 1 and dimension 2, demonstrating 80.28% data correlation). It was observed that, concerning CPO, the north region did not present a definite opinion, while the south and southeast regions strongly correlated with nonhealthy or moderately healthy opinions. The northeast did not have a definite opinion, while the midwest considered CPO to be slightly healthy (Figure 3a).

When evaluating RPO, the correlation analysis consisted of two dimensions (dimension 1 and dimension 2, demonstrating 87.96% data correlation). The northeast region perceived it to be slightly healthy, while the north region considered it moderately healthy. On the other hand, the southeast region classified RPO as healthy, while the south and midwest regions were neutral in their perception of RPO healthiness (Figure 3b).

### 3.6. Potential Purchase of Products with CPO and RPO

Regarding the likelihood of buying products with CPO, 541 respondents (50.8%) consider it very or extremely likely, while 323 individuals (30.3%) think it very or extremely likely to buy products with RPO (Figure 4).

Figure 5 presents the results of correspondence analyses based on the purchasing trend of products containing CPO or RPO (very likely, likely, neutral, unlikely, very unlikely) and the geographical location of respondents (north, northeast, south, midwest, southeast). It consists of two dimensions (dimension 1 and dimension 2, demonstrating 90.17% data correlation). Regarding products containing CPO, the north and northeast regions showed a greater inclination to buy them, correlating with the option “very likely”. In contrast, the south and midwest regions remained indifferent or less inclined, connecting with the responses “neutral” and “very unlikely”. The southeast region showed a moderate inclination to buy them by correlating with the option “likely” (Figure 5a).

The correspondence analysis of the purchasing trend of products containing RPO and the geographical location of respondents revealed two dimensions (dimension 1 and dimension 2), with 91.44% data correlation. The northeast region showed indifference or an unlikely tendency to buy products with RPO, while the midwest region also demonstrated this trend. The southern region was classified as very unlikely to purchase products with RPO, while the north and southeast were classified as very likely (Figure 5b).

## 4. Discussion

### 4.1. Geographic Distribution of Survey Participants

The survey revealed that most participants reside in the northeast region, likely due to its origin in Bahia and the traditional consumption of CPO in regional dishes [17,18]. The north and midwest regions had fewer participants, possibly due to communication difficulties through email and social media.

### 4.2. Oil Consumption Preferences among Respondents: Emphasis on Olive Oil and Soybean Oil

Among respondents, olive oil was found to be the most popular when evaluating the most consumed oils. Despite not being the most produced oil in Brazil, where olive cultivation is still in the consolidation process [19] and a significant portion of consumption is imported (104.179 million liters imported in 2019/2020) [20,21], people may be more concerned about health. Olive oil, rich in oleic acid and antioxidant compounds, is associated with a lower incidence of cardiovascular diseases (CVD) [22].

Regarding soybean oil, the second most consumed by survey participants, Brazil is the third largest producer of this oil globally, producing 9.967 million tons up to November 2022/23, with approximately 78.5% of this quantity allocated for domestic consumption [1].

The production of CPO in Brazil is concentrated in the states of Pará (82%) and Bahia (16%) [4]. Despite high production in the northern region, CPO consumption is predominant in Bahia and Africa [18]. This explains its appearance among the top five oils consumed by respondents, as the majority of survey participants were from the northeast region, where Bahia is located, known for its regional dishes incorporating palm oil (Appendix A). This region also represents the highest purchasers of CPO (Appendix A).

### 4.3. Factors Influencing Oil Choices: Price and Flavor Evaluation

In this study, it was found that price, along with taste, were the most relevant factors in choosing which oil to consume (Appendix A). In their study with German consumers, Hinkes and Christoph-Schulz [22] found that many people are price-sensitive when deciding whether to buy foods with sustainable palm oil. Similarly, Reardon, Padfield, and Salim [23], in their study on consumers in Malaysia, Singapore, and the UK, found that price was a determining factor in whether consumers would pay more for products with sustainable palm oil. Guadalupe et al. [16] also supported these findings, reporting that 55% of Peruvian consumers consider sensory properties, including taste, important in choosing a food product.

In our study, when correlating the most consumed type of oil, the respondent’s region, and income, we found that participants living in the midwest and north regions were more associated with lower income and the consumption of soybean and coconut oil. In contrast, oils generally with higher prices, such as olive oil, sunflower oil, and corn oil, were more associated with people residing in the southeast region and higher income (Figure 1). According to data from the Brazilian Institute of Geography and Statistics [24], the southeast region contributes to 51.9% of Brazil’s gross domestic product (GDP), which may explain the higher consumption of these oils in this region, while the midwest and north regions represent the minor contributions, with 10.4% and 6.3% of the GDP, respectively.

### 4.4. Respondents’ Knowledge about CPO

When asked if participants had heard about CPO, the majority (53%) answered no, possibly because this oil is more commonly known in Brazil as palm oil (Almeida et al., 2013), and only 39.5% of respondents were aware of this information (Appendix A).

CPO is a source of lipids, carotenoids (500–2000 mg/kg), and vitamin E (150–1500 mg/kg) [2]. In the survey, it can be observed that many people are aware that this oil is a source of the mentioned nutrients. Although CPO is not a source of vitamin C, vitamin K, proteins, and carbohydrates, more than half of the respondents indicated at least one of these nutrients (Appendix A).

CPO is the largest vegetable source of α- and β-carotene—highly bioavailable provitamin A carotenoids—and vitamin E. These nutrients have antioxidant activities and act as health promoters, studied in the prevention of various severe disorders such as cancer, cardiovascular diseases, Parkinson’s disease, Alzheimer’s disease, bone disorders, ocular issues, and nephrological disorders [25,26,27,28]. This information is unclear to most respondents, as they preferred not to agree/disagree with the second statement (Appendix A).

CPO is the most produced vegetable oil in the world, with 78.22 million tons produced in 2022–2023 [1]. However, most respondents are unaware of this, as 77.93% have no formed opinion or disagree with this information. It was observed that for all other statements about CPO, most people answered correctly, demonstrating good knowledge among the population (Appendix A). This may be related to the high level of education among most respondents.

Of the total respondents, 63.84% agreed with the sixth statement, which refers to sustainability in food production. There is a global discussion about CPO and sustainability because its expansion as a monoculture and high production trigger negative environmental impacts such as deforestation, increased greenhouse gas emissions, and loss of habitat for some animals, such as orangutans. It also impacts native communities in Southeast Asia [29,30,31]. In Brazil, companies in Pará, the largest CPO producer in the country, generally adhere to the Roundtable on Sustainable Palm Oil (RSPO) standards, where the opening of new areas in virgin forests is prohibited. Most palm plantations are in old pasture areas, reducing environmental damage (Appendix A) [32]. However, it is known that monoculture cultivation can compromise the environment by not considering the necessary period for biodiversity restoration. There are also reports, such as that of the quilombola community of Alto Acará/PA, noting degradations in their territory caused by the use of pesticides, which, with rainfall, end up contaminating rivers and streams, affecting activities such as bathing, cooking, and fishing [33]. Thus, in the rural quilombola community of Castanhalzinho, in the Eastern Amazon, the abandonment of residents was observed due to the transformations that oil palm monoculture caused in the community, including the appearance of venomous animals and mosquitoes [34]. Therefore, these authors verified the residents’ resistance in this region through actions to preserve their territory and the fight against mechanisms used by more powerful groups whose goal is their domination. Oil palm production in monoculture also caused problems for Indigenous communities in Kalimantan, leading to land fragmentation and migration [35]. Also, in Kalimantan, Potter [36] reported that the Dayaks resisted oil palm cultivation, refusing to give up their lands and preferring traditional activities.

This way, oil palm monoculture production must be conducted sustainably, considering environmental and social impacts. A balanced coexistence between the oil palm agroindustry, nature, and Indigenous peoples is crucial to ensure the preservation of ecosystems and the wellbeing of the involved communities.

### 4.5. Labeling, Storage, and Disposal of Crude Palm Oil: An Analysis of Current Consumer Practices

Regarding the purchase of CPO, Souza et al. [37] found in their study that CPO was often sold without labels, and half of the CPO samples with labels did not list the ingredients, unlike the findings in the present study. This can be explained by the fact that most people purchase in supermarkets, where labeling is mandatory [38] (Appendix A).

The presence of labels helps consumers make food choices according to dietary recommendations by providing information on the nutritional content of food products [39,40]. To facilitate consumer interpretation of quantitative nutrient declarations on food items, the National Health Surveillance Agency (ANVISA) approved, in 2020, a Front-of-Pack Nutrition Labeling (FOPNL) model through Board Resolution (RDC) number 429, which came into effect on 09/10/2022 [41,42].

Once opened, CPO is best preserved in the refrigerator, as storage at room temperature favors oxidative reactions, as Almeida et al. [43] verified. These authors also found that exposure to light intensifies these oxidation reactions, degrades carotenoids, and causes color changes in products. For this reason, olive oil, for example, is marketed in amber-colored bottles—which, according to the results, does not occur in the case of CPO. Thus, inadequacies were observed regarding the best way to store CPO: 45.21% of people find it exposed to light, 87.97% of respondents reported that they store the oil at room temperature, 78.39% buy it in transparent plastic bottles, and 56.35% of people buy it in transparent glass bottles. Moreover, only 18.93% of people who buy CPO usually store it in the refrigerator, while the majority (73.27%) report storing the oil in the cupboard (Appendix A).

The research shows that most people identify oil rancidity by its smell and texture. The rancid taste is caused by compounds such as aldehydes, ketones, alcohols, hydrocarbons, esters, furans, and lactones, resulting from the decomposition of hydroperoxides formed by the autoxidation of the oil, and reactions between unsaturated fatty acids and oxygen [43]. Light and heat can accelerate this deterioration, affecting the oil’s quality [43].

Of the total respondents, 31.85% reported disposing of used oil in the trash, which can harm the environment and cause contamination of soil, groundwater, beaches, rivers, lakes, streams, and pipeline blockages. Soap production is a sustainable alternative, as it avoids environmental damage and can be a source of income [44]. However, the lack of information or promotion by oil collection institutions may be the reason for improper disposal (Appendix A).

### 4.6. Respondents’ Preferences Regarding the Use of Crude Palm Oil in Cooking

This study also addressed respondents’ preferences regarding the two phases of CPO (olein and stearin) for cooking. Of the total respondents, 43.5% preferred using a mixture of both phases. Additionally, 33.90% chose based on better appearance and taste, and 27.51% based on better nutritional quality. Regarding the use of CPO, the majority (76.90%) stated they would use it for cooked dishes, such as moquecas, caruru, vatapá, and abará, while 58.50% mentioned using it for frying, such as acarajé, fish, and dumplings. The dishes mentioned are typical of Bahian cuisine, and the low linoleic acid content and the almost absence of linolenic acid provide relative stability to CPO in the preparation process [18]. Additionally, CPO is less prone to deterioration due to the presence of antioxidant compounds like β-carotene, tocopherols, and tocotrienols, making it a good option for cooking [45].

### 4.7. Frequency of Consumption of Foods Containing Crude and Refined Palm Oil

CPO, an excellent source of vitamin A precursor carotenoids, has been incorporated into meals in countries like India and West Africa to prevent and treat vitamin A deficiencies [30]. In the present study, an average of 57.6% of people had never consumed typical CPO-containing foods, possibly because dishes with CPO are typical of Bahia, such as moqueca and acarajé, and are not part of the local food culture in other states (Appendix A). This explains why the northeast region provided the most relevant responses, as it is more known and consumed in various culinary preparations there. In contrast, other regions are aware of it but do not consume it as much, given the regional culinary differences in Brazil.

Among those who reported consuming these dishes, the semiannual consumption frequency was mostly declared by participants, which could be related to festive periods (Holy Week, Cosme, and Damião) or vacation periods when tourists visit Bahia, usually between December and February.

The consumption of these foods, when accompanied by beverages, usually involves high-calorie drinks. As these foods already provide a high amount of energy, pairing them with a beverage increases the meal’s caloric value: on average, an acarajé (100 g, without accompaniments like caruru and vatapá) has 289 kcal and 2.89 kcal/g energy density (ED; average). When consumed with a 350 mL soda, providing 130 kcal and 0.37 kcal/g ED, the meal corresponds to 419 kcal, approximately 0.93 kcal/g ED (low ED) [46]. Additionally, the accompaniments usually consumed with acarajé (vatapá, caruru) contribute to an even higher energy content in the meal.

Regarding RPO, at least 20% of the survey participants reported consuming foods containing it daily. Daily consumption of foods containing RPO can pose health risks due to chloropropanols and glycidols produced during its processing [13,14], in addition to the loss of important nutrients such as carotenoids and vitamin E [8].

### 4.8. Respondents’ Knowledge Regarding RPO

In addition to consumption in its crude form, to meet the need for a clear and less strongly flavored oil, CPO undergoes the refining process [11]. This process includes stages that can be carried out by chemical or physical means. Still, it is preferably conducted physically due to advantages in terms of cost, processing time, and impact on human health [47]. Then, CPO undergoes bleaching, deodorization stages, and other specific treatments, depending on the refining technique applied. After this process, CPO is usually referred to as simply palm oil (PO) [48].

In Brazil, despite RPO being present in the formulation of various processed products, those declaring “palm oil” or “palm oil” in their ingredient lists are mostly imported foods. When evaluating European consumers, Hartmann et al. [49] observed that consumers in Sweden, Poland, and France considered products labeled “free of palm oil” healthier due to psychological factors such as a preference for naturalness.

The refining of vegetable oils can generate food-originating contaminants that, according to the International Agency for Research on Cancer (IARC), can pose a carcinogenic risk in humans. However, 79.62% of respondents did not express an opinion or disagreed with the first statement in Appendix A, demonstrating that, despite the majority having a good education, this information is not known to respondents. Additionally, it is known that most CPOs produced (approximately 80%) are destined for refining in the food industry [50,51]. Still, 67.89% of individuals did not express an opinion or disagreed with the second statement. Regarding the third statement, 63.94% of respondents did not express an opinion or disagreed, showing that Brazilian consumers do not mind the presence of RPO in their food, which aligns with the 69.48% of respondents who did not claim to observe the presence of RPO on labels (Appendix A).

### 4.9. Participants’ Cognition Regarding CPO and RPO

When asked how healthy respondents considered CPO, the majority declared it healthy or moderately healthy. However, less than half of the participants thought of RPO in this way. This reflects the participants’ good understanding of these oils, as they comprehend that refining CPO makes it somewhat less healthy. This process leads to a reduction in carotenoids, vitamin E, and sterols, as well as the formation of compounds that can pose health risks, such as chloropropanols like 3-MCPD, 2-MCPD, and glycidol.

The correlation analyses to assess Brazilians’ cognition levels regarding CPO and RPO showed that the south and southeast regions were more associated with unhealthy or moderately healthy opinions about CPO. This may be related to unfamiliarity with this food, which is more popular in the northeast region as part of regional culinary preparations [5]. Concerning RPO, the southeast region classified it as healthy. According to the Household Budget Survey (POF) 2017–2018, the acquisition of ultraprocessed foods in Brazil increases with family income. Furthermore, the south and southeast regions are the largest consumers of ultraprocessed foods, often containing RPO [24].

### 4.10. Potential to Purchase Products Containing CPO and RPO

Regarding the potential to buy foods containing CPO or RPO, the chances of consumption were higher for CPO, which is a positive finding since it is known that the amounts of carotenoids and vitamin E in RPO are lower than in CPO. CPO has 500–2000 mg/kg (CODEX) of carotenoids and 150–1500 mg/kg (CODEX) of vitamin E, while RPO has 500–700 mg/kg [52] and 150–1500 mg/kg [8] of carotenoids and vitamin E, respectively.

Respondents residing in the north and northeast regions showed a greater tendency to purchase products containing CPO, possibly because these regions are the largest CPO producers in the country [4]. Conversely, the correspondence analysis of the tendency to purchase products containing RPO and the geographical location of respondents showed that the north and southeast regions are the most likely to acquire products with RPO. This is possibly related to the southeast being the largest consumer of ultraprocessed foods in the country. However, it contradicts the fact that the north is the region that consumes the least ultraprocessed foods in Brazil [24].

## 5. Conclusions

Among the Brazilians who participated in this survey, the majority had never heard of CPO, which is better known in Brazil as dendê oil. Regarding the knowledge of the studied population about this oil, it was found that the nutrients provided by CPO are well perceived by people. However, they are unaware that it is the most consumed oil globally, and its nutrients are studied for disease prevention. Less than half of the respondents usually buy CPO, and among those who do, most typically purchase less than 500 mL of oil semiannually from supermarkets with proper labeling.

Similarly, RPO is unknown to the respondents, who are unaware that most of the CPO produced is destined for refining and that this processing produces substances with carcinogenic risk. The respondents also demonstrated not caring about the presence of RPO in their foods and did not claim to perceive its presence on labels. Throughout the research, good cognition of the participants about CPO and RPO was verified because they understood that refining CPO makes it somewhat less healthy. This aligns with the higher consumption potential of CPO, as their intention to buy, more significant than for RPO, reflects the respondents’ preference for consuming crude palm oil over refined.

Being a good source of nutrients that can contribute to health promotion, this study points to the need to increase knowledge and awareness among Brazilians, especially regarding the consumption of CPO. This can help them incorporate it into their diet consciously, preferably added to the end of cooked preparations or consumed raw, aiming to avoid the degradation of thermosensitive compounds, while also considering the pre-established daily limits for lipids and saturated fat (maximum of 35% of the total energy value of the diet from lipids, with up to 10% of the energy value of the diet from saturated fat.

## Figures and Tables

**Figure 1 foods-13-02923-f001:**
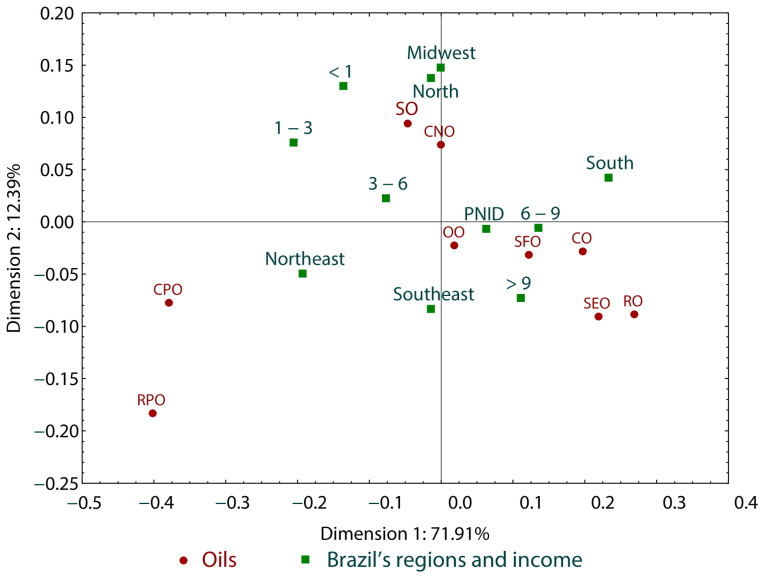
Correlation analysis between the most consumed type of oil, the participants’ region of residence, and their income. Note 1: SO = soybean oil, RPO = palm oil, OO = olive oil, CPO = crude palm oil, RO = canola oil, CO = corn oil, SFO = sunflower oil, SEO = sesame oil, CNO = coconut oil. Note 2: According to the exchange rate on USD 5.045 (https://economia.uol.com.br/cotacoes/cambio/, accessed on 1 March 2022), it was the equivalent to USD 240.219. Brazil’s national minimum salary in 2021 was BRL 1212.00.

**Figure 2 foods-13-02923-f002:**
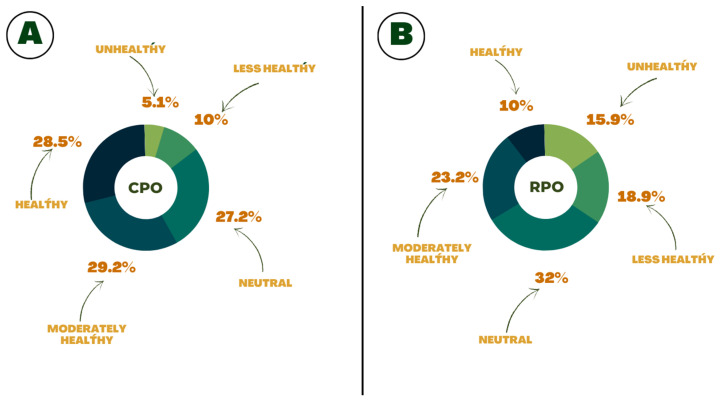
Respondents’ cognition about CPO (**A**) and RPO (**B**).

**Figure 3 foods-13-02923-f003:**
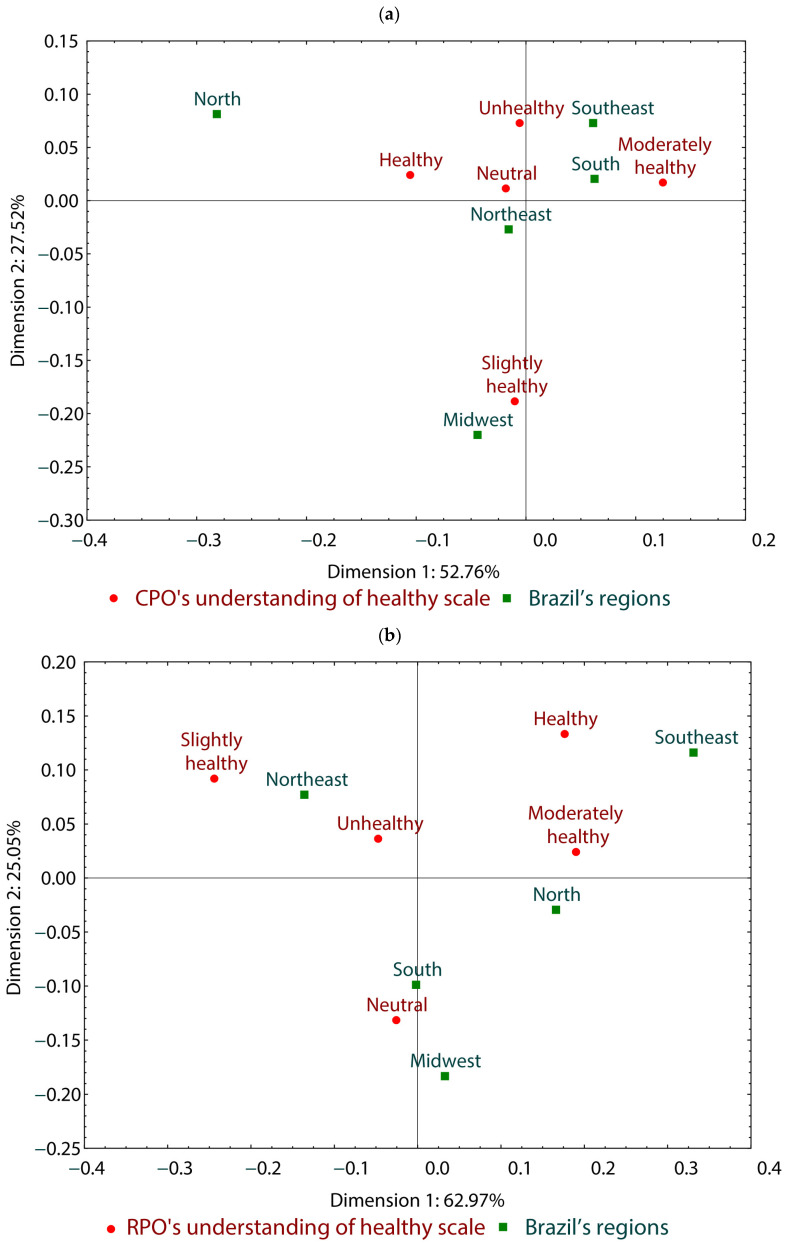
Correspondence analysis of understanding of healthiness of CPO (**a**) and RPO (**b**) by Brazil’s region in the survey participants.

**Figure 4 foods-13-02923-f004:**
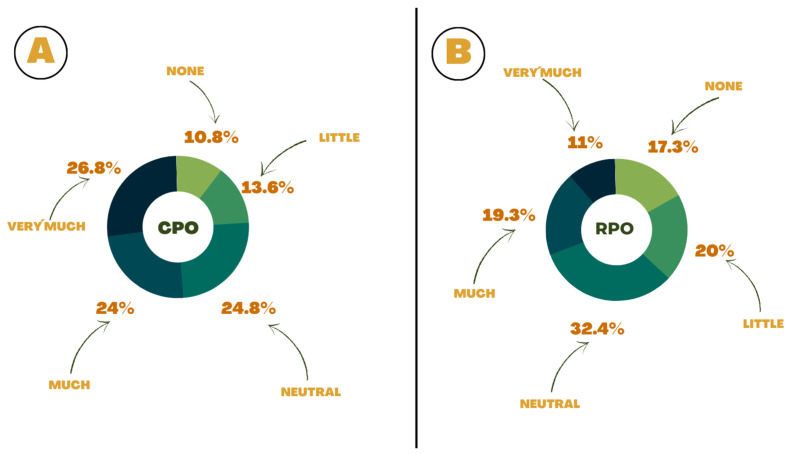
Likelihood of respondents purchasing products with CPO (**A**) or RPO (**B**).

**Figure 5 foods-13-02923-f005:**
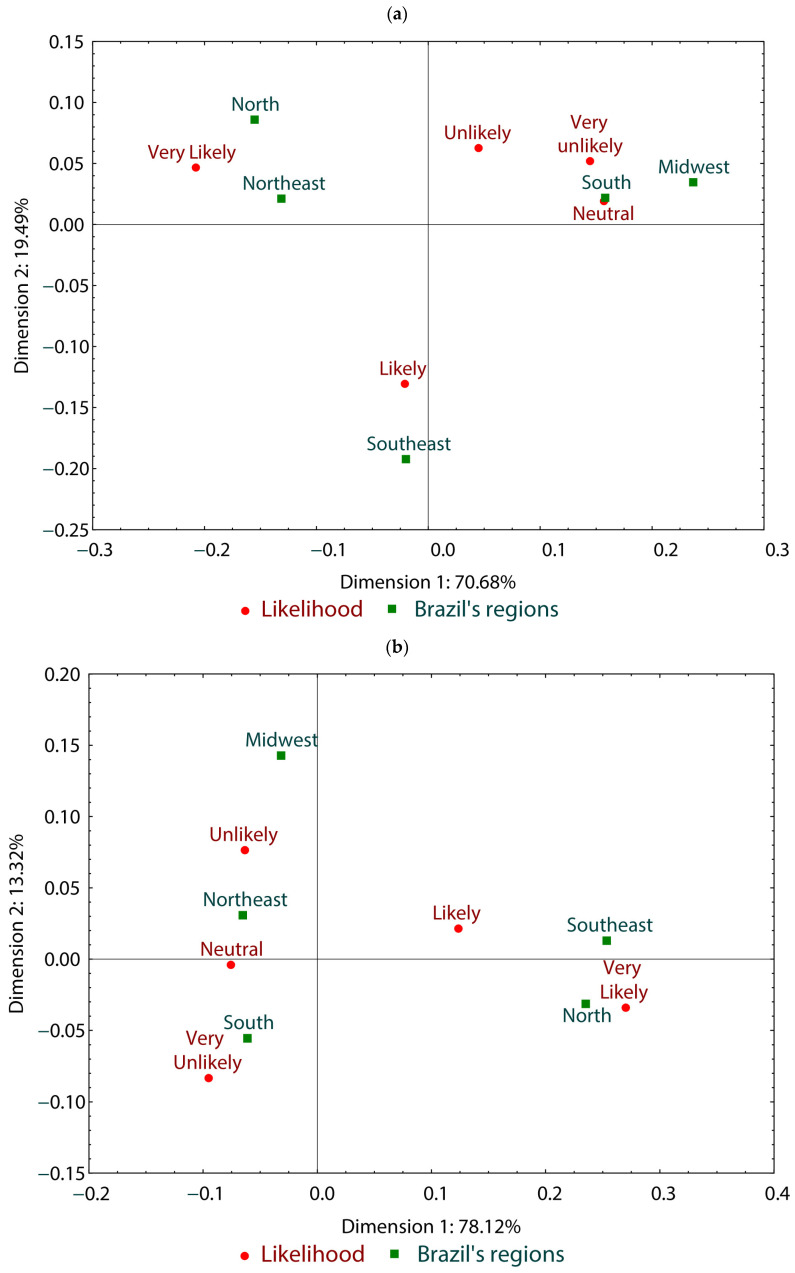
Analysis of correlation between the likelihood of purchasing foods containing CPO (**a**) or RPO (**b**) and regions in Brazil, based on survey data.

**Table 1 foods-13-02923-t001:** Description of the questions used in the research.

Topics	Questions
Socioeconomic aspects	Place of residence
	Gender identification
	Age group
	Number of people living with
	Level of education
	Occupation
	Family income
Preference	Types of oil
	Attributes for choice
	Flavor of CPO
	Fractions of CPO
Perception/Knowledge	Have you heard of CPO?
	Do you know it by another name?
	Opinions about CPO and RPO
	Nutrients that CPO is a source of
	Label information for CPO
	Perception of the ingredient RPO
	Foods that contain RPO in the composition
Attitude	Use of CPO in foods
Acquisition	Do you usually buy CPO?
	Where do you buy it?
	Quantity purchased
	How is the oil displayed?
	What is the packaging like?
	How do you perceive that the oil has gone bad?
	How do you dispose of it?
Frequency	Frequency of CPO purchases
	Frequency of consuming foods containing CPO or RPO
Cognition	How healthy does the consumer consider CPO and RPO to be?
Consumption potential	What is the likelihood of the consumer buying a product with CPO or RPO?

**Table 2 foods-13-02923-t002:** Sociodemographic profile of the participants.

Variable		N	%
Region of residence	Northeast	430	40.38
	South	368	34.56
	Southeast	134	12.58
	North	78	7.32
	Midwest	55	5.16
Gender identification	Female	743	69.77
	Male	317	29.76
	Prefer not to say	3	0.28
	Nonbinary	2	0.19
Age group	Between 18 and 29 years	275	25.82
	Between 30 and 39 years	293	27.51
	Between 40 and 49 years	266	24.98
	Between 50 and 59 years	160	15.02
	60 years or older	71	6.67
Number of people living in the household	1	135	12.68
	2	336	31.55
	3	294	27.60
	4	212	19.91
	5	68	6.38
	More than 5	20	1.88
Level of education	Postgraduate	671	63
	Completed undergraduate degree	146	13.71
	Completed high school	233	21.88
	Completed elementary school	8	0.75
	Incomplete elementary school	2	0.19
	Prefer not to answer	5	0.47
Occupation	Fulltime/Part-time employment	768	72.11
	Undergraduate/Graduate student	244	22.91
	Not working/Not studying	38	3.57
	Prefer to not answer	15	1.41
Approximate household income	Less than 1 minimum wage	53	4.98
	Between 1 and 3 minimum wages	206	19.34
	Between 3 and 6 minimum wages	200	18.78
	Between 6 and 9 minimum wages	163	15.30
	More than 9 minimum wages	382	35.87
	Prefer do not answer	61	5.73

Caption: N = number of respondents; % = percentage of responses relative to the total. Note: According to the exchange rate on USD 5.045 (https://economia.uol.com.br/cotacoes/cambio/, accessed on 1 March 2022), it was the equivalent to USD 240.219. Brazil’s national minimum salary in 2022 was BRL 1212.00.

**Table 3 foods-13-02923-t003:** Relationship between Brazilian states and the purchase of crude palm oil (n = 1065).

Regions	Answer Options	
	Purchase	Do Not Buy
North	34	44 (−)
Northeast	252 (+)	178 (−)
Midwest	21 (−)	34
Southeast	61	73 (−)
South	81 (−)	287 (+)

n = 1065. (+) or (−): Chi-square test indicates whether the observed value is higher or lower than the expected theoretical value (*p* < 0.05).

## Data Availability

The original contributions presented in the study are included in the article and Appendix A, further inquiries can be directed to the corresponding author.

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
