# Peer review of "Perception, Knowledge, and Consumption Potential of Crude and Refined Palm Oil in Brazilian Regions"

_foods, 2024, doi:10.3390/foods13182923_

Round 1

Reviewer 1 Report

Comments and Suggestions for Authors

This study aimed to comprehend the perception, knowledge, and consumption potential of crude and refined palm oil in Brazilian regions. This study is pioneering in the Brazil and could contribute to a better understanding Brazilian consumers regarding these oils.

A very interesting research was conducted, in this study.

The obtained data can be of great importance and practical application.

At the outset, I would draw attention to a few minor flaws.

The main title should not be in capital letters.

Author names, numbers indicating affiliation should be in superscript.

Elaeis guineensis – italics should be in the entire text of the manuscript.

Part 2. Material and Methods - 2.1. Participants: The study recruited participants (n=1,082). The authors should explain on what basis this number of recruited participants is sufficient for this kind of research.

Data were collected from March to July 2022 using the Google Form survey platform. A similar problem, whether and why this period was chosen and is it sufficient? What kind of data would there be in the second half of the year? What would the data be like in the same periods, but in different years? This requires some explanation. This is important so that this can be widely applied and reproduced.

Notes on Figure 2 should be rearranged.

1 - is confusing, as it is not clear whether it refers to Note 1, or to 1 - Olive oil.

It would be clearer if it said:

Note 1: Olive oils are extracted from the pulp of fruits, while oils are obtained from seeds, pits, or oleaginous grains (CARDOSO, 2006).

Note 2: In these questions, respondents could select all the alternatives they wished. Therefore, the sum of the percentages exceeded 100%.

The same applies to notes elsewhere in the manuscript (for example, Figure 3).

The titles of some figures are "Figure xx.", and some "Fig. xx." - this should be uniform.

A minor remark can be made about the color ratio in figures 1, 2 and 4. It would probably be better if the ratio of "dark" and "light" colors in these images were reversed - just as a suggestion.

All the figures on which the correspondence analysis is shown have poor quality (resolution) and the like. They show significant data, maybe it should be cleaner.

The conclusions are excellent, and fully correspond to the research objectives.

Finally, I express my great pleasure for the opportunity to review this manuscript, and I also commend the authors for the choice of the research topic, the way they conducted it and presented and analyzed the results, and gave the conclusions.

This is an authentic scientific and practical contribution. In support of this, there is also a very small percentage of similarities (13%).

Reviewer 2 Report

Comments and Suggestions for Authors

First and foremost, I would like to express my appreciation for the opportunity to review the manuscript titled "Perception, Knowledge, and Consumption Potential of Crude and Refined Palm Oil in Brazilian Regions," submitted for consideration to Foods.

After a thorough evaluation, I have carefully considered the relevance and suitability of the manuscript's theme in relation to the journal's scope and readership interests. It appears that the subject matter, while undoubtedly significant and insightful within its field, may not align perfectly with the thematic focus and editorial direction that the journal aims to uphold. The journal primarily seeks contributions that are closely related to Food sciences and technology,Food chemistry and physical properties,Food engineering and production,Food microbiology and safety,Food security and sustainability,Food toxicology,Sensory and food quality,Food analysis,Functional foods, food and health,Food psychology,Food and environment, and the current manuscript, although of high quality, seems to diverge from this specific thematic trajectory.

Additionally, the manuscript discusses the consumption potential of Crude Palm Oil (CPO) in the Brazilian market. It is important to note that, from a regulatory standpoint, CPO may not meet the criteria for direct sale to consumers in many jurisdictions due to the potential health risks associated with unrefined oils. While the manuscript provides valuable insights into consumer perceptions and knowledge regarding CPO, the practical application of these findings in the market may be limited by the need to comply with food safety and labeling regulations.

In light of these considerations, I regret to inform you that I cannot recommend the manuscript for publication in its current form. I encourage the authors to consider revising the manuscript to better align with the specific thematic focus of the journal or to seek publication in a journal that specializes in the broader aspects of food science and consumer behavior, where the manuscript's findings may find a more receptive audience.

I hope that this feedback is constructive and wish the authors every success in finding a suitable outlet for their important research.

Comments on the Quality of English Language

No answer.

Reviewer 3 Report

Comments and Suggestions for Authors

The manuscript (foods-3163744) entitled “Perception, knowledge, and consumption potential of crude and refined palm oil in Brazilian regions” presents interesting findings from a survey of perception and knowledge about palm oil. The manuscript is well-structured, the survey was correctly designed, and the results clearly described and discussed. However, some overall recommendations are suggested as follows:

 Abstract. Provide more findings of the study.

Page 8. Rewrite the phrase “About the second statement about the study of its antioxidant substances…” Check and modify the preposition “about”.

L124. Define OPB the first time.

Page 9, 14, 15 and L24, 31, 59, 75, 153, 189, 201, 229. The CPO abbreviature was already defined at the beginning of the manuscript.

Page 14, 15 and L26, 31, 49, 222, 235. The RPO abbreviature was already defined at the beginning of the manuscript.

L234. The PO abbreviature was already defined at the beginning of the manuscript.

L252. Define CPB the first time. Probably, is it a typing error? Revise the next sections.

L264. Define POF the first time.

L304. Suppress or define TEV the first time.

Round 2

Reviewer 2 Report

Comments and Suggestions for Authors

no comments

Author Response

I want to extend my sincere gratitude for your invaluable contributions and insightful feedback on our manuscript. Your detailed comments and suggestions have greatly enhanced the quality of our work, and we sincerely appreciate the time and effort you dedicated to reviewing it.

Your expertise has been instrumental in refining our arguments and improving the overall presentation of our research. We are confident that the revisions will significantly strengthen the manuscript, and we are grateful for your guidance throughout this process.

Again, Thank you for your support and for helping us advance our work. We look forward to the opportunity to contribute further to the field through your esteemed journal.